# Generation of Human Regulatory Dendritic Cells from Cryopreserved Healthy Donor Cells and Hematopoietic Stem Cell Transplant Recipients

**DOI:** 10.3390/cells12192372

**Published:** 2023-09-28

**Authors:** Sabrina M. Scroggins, Annette J. Schlueter

**Affiliations:** 1Department of Biomedical Sciences, University of Minnesota-Duluth, 1035 University Drive, 341 SMED, Duluth, MN 55812, USA; 2Department of Pathology, University of Iowa, Iowa City, IA 52242, USA; annette-sclueter@uiowa.ecu

**Keywords:** good manufacturing practices (GMPs), graft versus host disease (GVHD), hematopoietic stem-cell transplant (HSCT), monocytes, regulatory dendritic cells (DCreg), human

## Abstract

Acute graft versus host disease (GVHD) remains a significant complication following hematopoietic stem cell transplant (HSCT), despite improved human leukocyte antigen (HLA) matching and advances in prophylactic treatment regimens. Previous studies have shown promising results for future regulatory dendritic cell (DCreg) therapies in the amelioration of GVHD. This study evaluates the effects of cryopreservation on the generation of DCreg, the generation of young and older DCreg in serum-free media, and the feasibility of generating DCreg from young and older HSCT patient monocytes. DCregs were generated in X-vivo 15 serum-free media from donor or patient monocytes. This study includes the use of monocytes from young and older healthy, donor, and HSCT patients with varying hematological diseases. Phenotypic differences in cell populations were assessed via flow cytometry while pro-inflammatory and anti-inflammatory cytokine production was evaluated in culture medium. The number of DCreg generated from cryopreserved monocytes of healthy donors was not significantly different from freshly isolated monocytes. DCreg generated from cryopreserved monocytes had comparable levels of co-stimulatory molecule expression, inhibitory molecule expression, and cytokine production as freshly isolated monocytes. Young and older healthy donor monocytes generated similar numbers of DCreg with similar cytokine production and phenotype. Although monocytes from older HSCT patients generated significantly fewer DCreg, DCreg from young and older HSCT patients had comparable phenotypes and cytokine production. Monocytes from young and older myelodysplastic syndrome (MDS) patients generated reduced numbers of DCreg compared to non-MDS-derived DCreg. We demonstrate that the cryopreservation of monocytes from HSCT patients of varying hematological diseases allows for the cost-effective generation of DCreg on an as-needed basis. Although the generation of DCreg from MDS patients requires further assessment, these data support the possibility of *in vitro*-generated DCreg as a therapy to reduce GVHD-associated morbidity and mortality in young and older HSCT recipients.

## 1. Introduction

Acute graft versus host disease (GVHD) is a systemic immune-mediated disease primarily involving the skin, liver, and gastrointestinal tract [1,2,3,4]. Despite improved human leukocyte antigen (*HLA*) matching and advances in prophylactic treatment regimens, GVHD following hematopoietic stem cell transplant (HSCT) remains a significant complication. GVHD occurs in 40–85% of young and up to 70% of older allogeneic transplant recipients and is fatal in 30–53% of those patients [5,6,7,8,9,10]. To this end, age is an independent risk factor for increased incidence and severity of GVHD [4,11,12,13]. The number of patients over 60 undergoing transplants each year has steadily increased over the last decade [9], making up 41% of allogeneic transplants [14]. As the total number of HSCTs performed each year is steadily increasing, so is the need for alternative therapies for GVHD in both young and older HSCT recipients.

In a murine model of GVHD, a single treatment with age-matched, syngeneic regulatory dendritic cells (DCregs) in mice attenuated allogeneic-bone marrow transplant (BMT)-induced GVHD [15,16] while maintaining the graft versus tumor (GVT) response [17,18]. Importantly, our studies also demonstrate the amelioration of GVHD in older mice with the administration of DCregs following allogeneic BMT [16]. Human DCregs generated from young, healthy donor peripheral monocytes induces regulatory T cells (Tregs) and T cell anergy *in vitro* [15,19]. These DCregs, however, were generated in fetal bovine serum (FBS)-containing media that potentially exposes the cells to animal immunogens. Additionally, monocytes for DCreg generation must be obtained from HSCT patients prior to receipt of transplant, thus requiring cryopreservation to allow for post-transplant treatment with DCregs. To be more clinically relevant and for future translational evaluation, the feasibility of generating DCregs in bovine serum-free media from young and older HSCT recipients, as well as the effects of cryopreservation on DCreg generation, need to be considered.

## 2. Materials and Methods

### 2.1. Mononuclear Cell Sources

All samples and informed consent were obtained at the University of Iowa Hospitals and Clinics from “young” (age 30 or below) and “older” (age 50 or above) individuals in accordance with approved IRB 200604728. A coded identification number was assigned to each sample as obtained. Healthy young and older donor monocytes were purified from de-identified leukoreduction system (LRS) cones obtained from the DeGowin Blood Center, Iowa City, IA, USA. Peripheral blood (PB) samples were obtained from patients upon admission to the hospital, directly prior to HSCT for varying hematological diseases. These hematological diseases included acute lymphoblastic leukemia (ALL), acute myeloid leukemia (AML), chronic lymphocytic leukemia (CLL), chronic myeloid leukemia (CML), non-Hodgkin lymphoma (NHL), myelodysplastic syndrome (MDS), and myelofibrosis. HSCT patients were not in remission at the time of sample acquisition.

### 2.2. Cell Preparation

Dulbecco’s phosphate-buffered saline (PBS) without calcium, magnesium, or phenol red (Thermo Fisher Scientific, Waltham, MA, USA) was used to flush LRS cones prior to monocyte enrichment. Prior to transplant, 20–30 mL of whole blood in EDTA was obtained from HSCT recipients and washed with sterile PBS. Following washing, cells from healthy donor and HSCT recipients were labeled with Rosettesep human monocyte enrichment cocktail (Stemcell Technologies, Cambridge, MA, USA) per the manufacturer’s protocol and centrifuged through sterile, endotoxin-free Histopaque^®^-1077 gradient medium per the manufacturer’s guidelines (Sigma-Aldrich, St. Louis, MO, USA). Monocytes were recovered from the interface between the layer of PBS and Histopaque^®^-1077.

### 2.3. Cryopreservation of Monocytes

Monocytes obtained from healthy donor LRS cones were cryopreserved at 5 × 10^7^ cells/mL in X-vivo serum-free media (Lonza, Basel, Switzerland) containing sterile 10% dimethylsulfoxide (DMSO; Sigma-Aldrich, St. Louis, MO, USA). Cells were stored in a −80 °C freezer for 48 h followed by transfer into liquid nitrogen until use (3 weeks–12 months).

### 2.4. DCreg and Conventional Dendritic Cell (cDC) Cultures

Freshly isolated or cryopreserved monocytes were differentiated into DCregs similarly to those previously described [15,16] in serum-free media [15]. Briefly, 1 × 10^6^ cells/mL were cultured in X-vivo 15 media supplemented with human transforming growth factor beta 1 (TGFβ), granulocyte macrophage-colony stimulating factor (GM-CSF), interleukin 4 (IL-4), and IL-10 (Peprotech, Rocky Hill, NJ, USA; 50 ng/mL each), for 10 days. Human tumor necrosis factor alpha (TNFα; Sigma-Aldrich, St. Louis, MO, USA; 1 µg/mL) was added to the cultures for the last 3 days of culture. cDC were generated in the same manner, but only human GM-CSF and IL-4 cytokines were used, followed by TNFα.

### 2.5. Flow Cytometric Reagents

To prevent non-specific FcγR binding, all cell samples were incubated with anti-CD16/32 (Thermo Fisher Scientific, Waltham, MA, USA). Fluorochrome-conjugated, purified mouse immunoglobulins were used as isotype controls for background fluorescence. Cells were stained with the following fluorochrome-conjugated monoclonal antibodies (mAbs): CD11c (clone 3.9, PE-Cyanine7), HLA-DR (clone L243, Alexa Fluor^TM^ 488), CD40 (clone 5C3, PE), CD45RO (clone UCHL1, PE), CD80 (clone 2D10.4, Super Bright 600), CD86 (clone IT2.2, Alexa Fluor^TM^ 700), ILT-3 (clone ZM4.1, APC), CD25 (clone BC96, eFluor^TM^ 450), CD103 (clone B-Ly7, PE), and Foxp3 (clone PCH101, APC) (Thermo Fisher Scientific, Waltham, MA, USA), and PD-L1 (clone 29E.2A3, Brilliant Violet 650^TM^) (Biolegend, San Diego, CA, USA). Following staining, cells were fixed in 2% formaldehyde (pH 7.4) until acquisition.

### 2.6. Flow Cytometric Staining and Analysis

Flow cytometric data were obtained within 48 h on a Becton Dickson FACSCanto II or LSR II (San Jose, CA, USA) and analyzed using FlowJo software (FlowJo v10.9.0, LLC, Ashland, OR, USA). Dead cells were excluded via forward/orthogonal light scatter characteristics. Single cells were identified by forward scatter area versus side scatter width. Live cells were gated and DCs were identified as HLA-DR+ CD11c+.

### 2.7. Culture Cyotokine Assessment

At the end of the culture, DCreg and the cDC culture medium were harvested and stored at −80 °C until analysis. Each culture condition was assayed in triplicate using human/mouse TGFβ1 (Biolegend, San Diego, CA, USA), human IL-6, IL-10, IL-12p70, and TNFα Ready-Set-Go ELISA kits per the manufacturer’s protocols (ThermoFisher, Waltham, MA, USA).

### 2.8. Treg Induction Assay

As described above, DCreg was generated from donor 1 (Appendix A). On day 10 of DCreg culture, CD4+ T cells were purified from a donor 2 LRS cone using a Rosettesep human CD4+ enrichment cocktail per the manufacturer’s guidelines (Stemcell Technologies, Cambridge, MA, USA; Appendix A). 5 × 10^6^ CD4+ T cells from donor 2 were then co-cultured with 5 × 10^5^ DCreg from donor 1 in X-vivo media for 5 days. Cells were stained with mAbs to CD45RO and Foxp3 to assess the induction of Treg (Appendix A). Treg were defined as Foxp3+ cells.

### 2.9. Live Cell Imaging

Images of HSCT patient DCreg were taken at the end of culture, prior to flow cytometric staining. Images were acquired on a Nikon TS100 phase-contrast inverted microscope equipped with a Retiga 2000R digital camera at 40× magnification.

### 2.10. Statistical Analysis

Statistical significance was determined using a two-tailed Student *t*-test or one-way ANOVA and Tukey–Kramer Multiple Comparisons test, where appropriate (GraphPad, Prism 10, La Jolla, CA, USA). Statistical significance was designated at α = 0.05 or as determined by Bonferroni correction for multiple comparisons ANOVA.

## 3. Results

### 3.1. Compared to cDC, DCreg Express Lower Levels of CD11c and HLA-DR

Because not all HSCT recipients require GVHD, the most cost-effective means of treatment would be to generate DCregs on an as-needed basis in a personalized medicine approach. Thus, monocytes need to be cryopreserved prior to HSCT for the later generation of DCregs. Unlike murine DCregs [16], the generation of human DCregs from peripheral monocytes yields a double-positive (HLA-DR+ and CD11c+) DC population (Appendix A). This is likely due to a difference in the cellular source, as murine DCregs are generated from bone marrow and not peripheral blood. When compared to cDCs, young and older DCregs display reduced expression of CD11c and HLA-DR.

### 3.2. Cryopreservation of Young, Healthy Monocytes Prior to DCreg Generation Does Not Negatively Impact Cell Number, Phenotype, or Cytokine Production

To assess the effect of cryopreservation, monocytes were subdivided into two groups, one for fresh (immediate) DCreg generation and the other for cryopreservation and subsequent DCreg generation. For phenotypic analysis and cytokine production evaluation, DCregs were generated from either freshly isolated or cryopreserved monocytes in X-vivo. The number of DCregs generated in culture following cryopreservation did not significantly differ from the number of DCregs generated from freshly isolated and cultured monocytes (Figure 1A). Furthermore, DCregs generated from cryopreserved monocytes had similar levels overall of co-stimulatory (Figure 1B) and co-inhibitory molecule (Figure 1C) expression as DCregs generated from freshly isolated monocytes. The only exception is the inhibitory molecule CD103, which was increased on cryopreserved DCregs (Figure 1C). The secretion of TNFα, IL-10, and TGFβ production by DCregs were comparable between both groups (Figure 1D,E), suggesting DCreg function was not significantly affected by cryopreservation. IL-6 production, however, was significantly reduced following cryopreservation (Figure 1D). Given that overall DCreg surface molecule expression and cytokine production were not altered by cryopreservation of monocytes prior to DCreg generation, these data support cryopreservation for the generation of DCregs on an as-needed basis.

### 3.3. DCregs Generated from Older, Healthy Donors Have Comparable Numbers, Phenotype, and Function to DCregs Generated from Young, Healthy Donors

DCregs were generated in X-vivo 15 serum-free media from young and older donors, freshly isolated or cryopreserved monocytes, and evaluated for phenotype and function. As shown in Figure 2A, monocytes from young and older healthy donors generated comparable numbers of DCregs. Phenotypic differences between cDCs and DCregs generated from young and older healthy donors were assessed via flow cytometry. As expected, young and older cDCs highly expressed the co-stimulatory molecules CD80 and CD40 (Figure 2B). Conversely, DCregs had very low expression of CD80 and CD40 compared to cDCs, regardless of donor age (Figure 2B). Additionally, DCregs had significantly higher expression of the co-inhibitory molecules ILT3, PD-L1, CD103, and CD25 compared to cDCs (Figure 2C). Importantly, older DCregs were phenotypically similar to DCregs generated from young donors (Figure 2B,C). There was no statistical difference in surface molecule expression between young and older donors in DC populations. The minimal co-stimulatory molecule expression with a high level of inhibitory molecule expression on human DCregs is comparable to the phenotype observed in murine DCregs with a tolerogenic function *in vivo*.

As an indicator of function, cDC and DCreg pro-inflammatory and anti-inflammatory cytokine production was evaluated in the culture medium at the end of the culture. In agreement with previous reports [20], cDCs, as well as DCregs, generated from monocytes in X-vivo media did not produce IL-12p70.) Young and older cDCs and DCregs produced comparable amounts of TNFα, while IL-6 production was significantly increased from DCregs (Figure 2D). Like murine DCregs, human DCregs but not cDCs produced significant amounts of IL-10 and TGFβ (Figure 2E). The production of cytokines did not differ between young and older donors. Interestingly, one young and one older DCreg culture were excluded from the IL-10 analysis as they had IL-10 concentrations exceeding the detection limit of the assay (exceeded 2 × 10^5^ ng/mL) and required further dilution to determine the concentration.

### 3.4. Monocytes from Patients with Various Hematological Diseases Generate Phenotypical and Functional DCregs

Hematopoietic disease may negatively impact the generation of DCregs from the monocytes of HSCT patients. To address this possibility, 56 HSCT patients were enrolled, 12 young and 44 older patients (Appendix A). Acute lymphocytic leukemia (ALL) was the most represented disease in young patients. Acute myeloid leukemia (AML), chronic myeloid leukemia (CML), and non-Hodgkin lymphoma (NHL) patients were represented in both young and older populations. Although myelodysplastic syndrome (MDS) primarily occurs in older patients, occurring in 12 of the older patients in our study, MDS was observed in 1 young patient. Myelofibrosis and chronic lymphocytic leukemia (CLL) patients were only represented in the older population.

Like DCregs generated from healthy donors, DCregs generated from young and older HSCT patients expressed HLA-DR and CD11c (Appendix A). Both monocytes from young and older HSCT patients generated DCregs; however, older patients’ monocytes produced significantly fewer DCregs (Figure 3A). Consistent with murine DCregs, young and older patient DCregs expressed similar levels of co-stimulatory (Figure 3B) and co-inhibitory molecules (Figure 3C). Although patient DCregs had low to moderate expression of the co-stimulatory molecules CD80 and CD40 (Figure 3B), they highly expressed PD-L1 and ILT3 (Figure 3C). Additionally, patient DCregs expressed low levels of CD103 and CD25 (Figure 3C). The production of pro-inflammatory (IL-6 and TNFα) (Figure 3D) and anti-inflammatory (IL-10 and TGFβ) (Figure 3E) cytokines were also comparable between young and older patient DCregs. The production of IL-10 was heterogeneous between patients. Three of the young and seven of the older patients’ DCregs had IL-10 concentrations exceeding 2 × 10^5^ ng/mL. As with DCregs generated from healthy donors, patient DCregs did not produce IL-12p70. Monocytes from older MDS patients generated significantly reduced numbers of DCregs compared to non-MDS monocytes (Figure 3F). Although statistics on young MDS could not be performed (*n* = 1), monocytes from the young MDS patient generated fewer DCregs than young non-MDS monocytes (Figure 3F). Interestingly, DCreg cultures from young ALL and AML, as well as older NHL patients, had both uniformly round and elongated spindle-shaped cells (Appendix A). Conversely, cultures generated with monocytes from patients with MDS, regardless of the treatment regimen, had very few cells and large masses of debris (Appendix A). These findings suggest further investigation into better methods for generating DCregs from MDS patients is required. It is possible that MDS patients are not good candidates for DCreg treatment due to the inability to efficiently generate DCregs from these individuals.

DCreg function was further confirmed through the ability of DCregs to induce Treg. On day 10 of the DCreg culture (donor 1), naïve CD4+ T cells were purified using a CD4+ Rosettesep enrichment cocktail from a second LRS cone (donor 2) (Appendix A). CD4+ T cells were either placed in the culture alone or in the presence of DCregs at a 10:1 ratio for 5 days and then stained for Foxp3 via intranuclear staining. Approximately 5% of the CD4+ T cells were Foxp3+ at baseline, as indicated in the T-cell-only cultures (Figure 3G). Confirming DCreg function in cells generated in serum-free media, a higher frequency of Treg cells were present in cultures containing DCregs compared to T-cell-only cultures (Figure 3G). Additionally, this equated to a numerical difference of approximately 5 × 10^4^ Tregs per 1 × 10^6^ T cells cultured, compared to 2 × 10^4^ Tregs per 1 × 10^6^ T cells in the T-cell-only cultures. Taken together, these data indicate DCregs generated from young and older healthy donors, in serum-free media, are phenotypically and functionally tolerogenic.

## 4. Discussion

DCregs generated from young, syngeneic BM has been demonstrated to alleviate GVHD-induced mortality in young mice [15,16]; importantly, the GVT response remained intact following DCreg treatment [17,18]. Further, we previously demonstrated that DCregs generated from older mice reduces GVHD-associated morbidity and mortality in older BMT recipient mice [16]. Recent work in rats has shown that DCregs alleviate autoimmune disease via the formation of microchimerism in treated rats [21]. These data provide support for the potential of DCreg therapy in the long-term amelioration of human GVHD. Previous studies have demonstrated that DCs with low co-stimulatory molecule expression and the ability to induce Tregs and T cell anergy *in vitro* can be generated from young, healthy donor monocytes in FBS-containing media [15]. Further, treatment with anti-inflammatory agents or immunosuppressive drugs has been shown to result in DCreg differentiation [22]. The use of animal serum is not compliant with good manufacturing practices (GMPs) and, to our knowledge, DCreg generation from older individuals or from HSCT patients has not been addressed. Also, to be clinically relevant and economically feasible, monocytes would need to be cryopreserved for the generation of DCregs on a case-by-case basis for a personalized medicine approach. Cryopreserved monocytes from young and older healthy donors generated similar numbers of DCregs as freshly isolated monocytes. DCregs generated from cryopreserved monocytes maintained high expression of co-inhibitory molecules and anti-inflammatory cytokine production. There were no age-dependent variations between cryopreserved and freshly isolated monocytes. These data suggest that DCregs generated from cryopreserved monocytes are phenotypically and functionally comparable to those generated from freshly isolated monocytes. Additional confirmation of this function may be evaluated via the induction of T cell anergy in future studies.

This study demonstrates that DCregs can be generated in serum-free media from young and older healthy donors and HSCT patient monocytes. Although DCreg generation from young and older healthy monocytes generated similar numbers of DCregs, older-patient monocytes generated fewer DCregs per cultured monocyte compared to young-patient monocytes. Due to the lower efficiency, future studies may focus on developing protocols for the expansion of DCregs *in vitro.*

In agreement with DCregs generated in serum-containing media [15], DCregs generated in X-vivo serum-free media from healthy donors and HSCT patients maintained low co-stimulatory molecule expression. Expression of CD25, CD103, PD-L1, and ILT3 have on various DC populations and are thought to play a role in regulating immune activation and tolerance [21,23,24,25,26,27,28,29,30,31,32,33]. Human DCs that inhibited T cell function were shown to co-express CD25 and indoleamine 2,3-dioxygenase (IDO) [27]. Indeed, all human DCregs generated in this study expressed CD25. As murine DCregs also produced IDO and were shown to be necessary for maximal protection from GVHD, it would be of interest to evaluate human DCreg IDO expression. We have previously demonstrated that PD-L1 and PIR B expression on DCregs is required for protection from GVHD in BMT mice [16]. Human healthy donor and patient DCregs generated in X-vivo media also highly expressed PD-L1 and ILT3.s

IL-10 and TGFβ are anti-inflammatory cytokines involved in Treg function; TGFβ has been shown to be critical in the development and maintenance of Tregs [34,35]. In accordance with previous reports describing DC with regulatory functions [29,36,37,38] and our murine DCregs [16], DCregs generated *in vitro* from healthy and HSCT patient monocytes produced substantial amounts of IL-10 and TGFβ. Confirming human DCreg function and in accordance with previous reports describing DCregs that induce Tregs and T cell anergy [15,17,29,38,39,40,41,42], human DCregs generated in X-vivo induced Tregs *in vitro*. The induction of T cell anergy by human DCregs was not evaluated in the current study. Although the requirement for specific co-inhibitory molecules and/or anti-inflammatory cytokines was outside the scope of the current study, the phenotypic and functional data suggest human DCregs and murine DCregs may function through similar mechanisms.

For the first time, these data demonstrate that DCregs generated under clinically relevant conditions (i.e., in serum-free media and from cryopreserved monocytes) have a similar phenotype and function as DCregs generated from freshly isolated and cultured monocytes. Additionally, DCregs can be generated from older healthy donors with similar numbers, phenotype, and functions to young donors. Perhaps most importantly, DCregs can be reliably generated from young and older HSCT patients with a wide variety of hematological diseases. These DCregs express comparable co-inhibitory surface molecules and produce anti-inflammatory cytokines as DCregs generated from healthy donor monocytes. Many of the DCreg cultures from MDS patients (5/12 older patients and 1/1 young patient) had considerable debris and very low cell recovery, preventing phenotypic analysis and cellular yield inclusion. These findings suggest further investigation into alternative methods for generating DCregs from MDS patients is required. It is possible that MDS patients are not good candidates for DCreg treatment due to the inherent defects in monocytes resulting in an inability to efficiently generate DCregs from these individuals. A larger, more comprehensive study is warranted to clarify the impact of specific hematological diseases and disease states on the generation and function of DCregs.

## 5. Conclusions

Our study demonstrates that DCregs can be generated under clinically relevant conditions from both young and older individuals. Further, to increase personalization and feasibility, DCregs can be generated following the cryopreservation of monocytes on an as-needed basis for a personalized medicine approach to treat GVHD. Lastly, DCregs generated from peripheral monocytes of HSCT patients prior to transplant are phenotypically and functionally active. Collectively, our study provides critical data to support the use of *in vitro*-generated DCregs as a therapy to reduce GVHD-associated morbidity and mortality in HSCT recipients.

## Figures and Tables

**Figure 1 cells-12-02372-f001:**
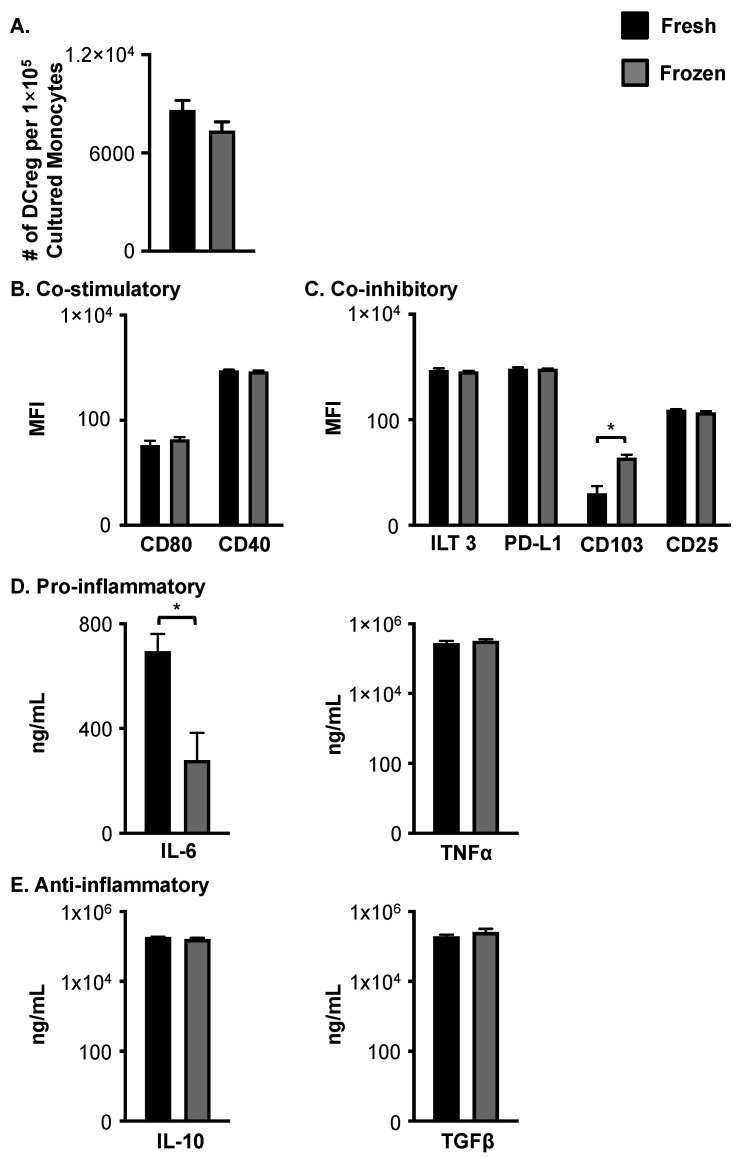
Cryopreservation does not significantly impact the generation, phenotype, or cytokine production of DCregs from healthy donors. (**A**) Freshly isolated and cryopreserved monocytes from young, healthy donors generated comparable numbers of DCregs per 1 × 10^5^ cultured monocytes. *N* = 8 per group. DCregs generated from young, freshly isolated, and cryopreserved monocytes have a similar surface phenotype, whereby they express (**B**) low levels of co-stimulatory molecules and (**C**) high levels of inhibitory molecules on their surface. *N* = 4 per age group. (**D**) DCregs generated from young, healthy cryopreserved monocytes secreted less pro-inflammatory IL-6, while (**E**) production of anti-inflammatory cytokines is not impacted by cryopreservation. *N* = 11 per group. Data are mean ± SEM.* = *p* < 0.05.

**Figure 2 cells-12-02372-f002:**
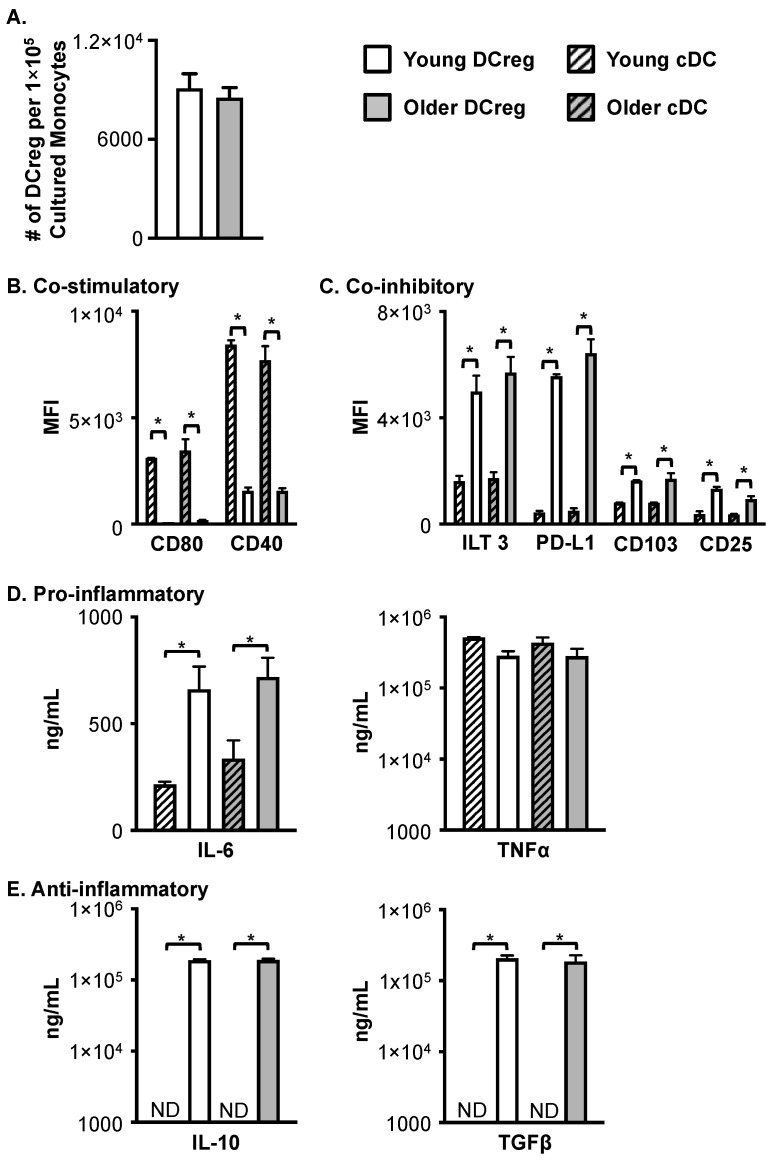
Successful generation of DCregs from young and older healthy donors under clinically relevant conditions. (**A**) DCregs generated from young and older healthy donor monocytes have a similar number of DCregs per 1 × 10^5^ cultured monocytes. *N* ≥ 8 donors per age group. Young and older healthy donor generated DCregs have a similar phenotype in that they express (**B**) low levels of co-stimulatory molecules and (**C**) high levels of inhibitory molecules. *N* = 4 per age group. DCregs generated under clinically relevant conditions from young and older monocytes secrete similar levels of pro- (**D**) and anti- (**E**) inflammatory cytokines. ND, not detected. IL-6, TNFα, and TGFβ ELISAs *n* = 4 young; *n* = 4 older. IL-10 production exceeded maximal standard in 1 young and 1 older DCreg culture (exceeded 2 × 10^5^ ng/mL IL-10) and was not included (*n* = 3 young; *n* = 3). Data are mean ± SEM. * = *p* < 0.05.

**Figure 3 cells-12-02372-f003:**
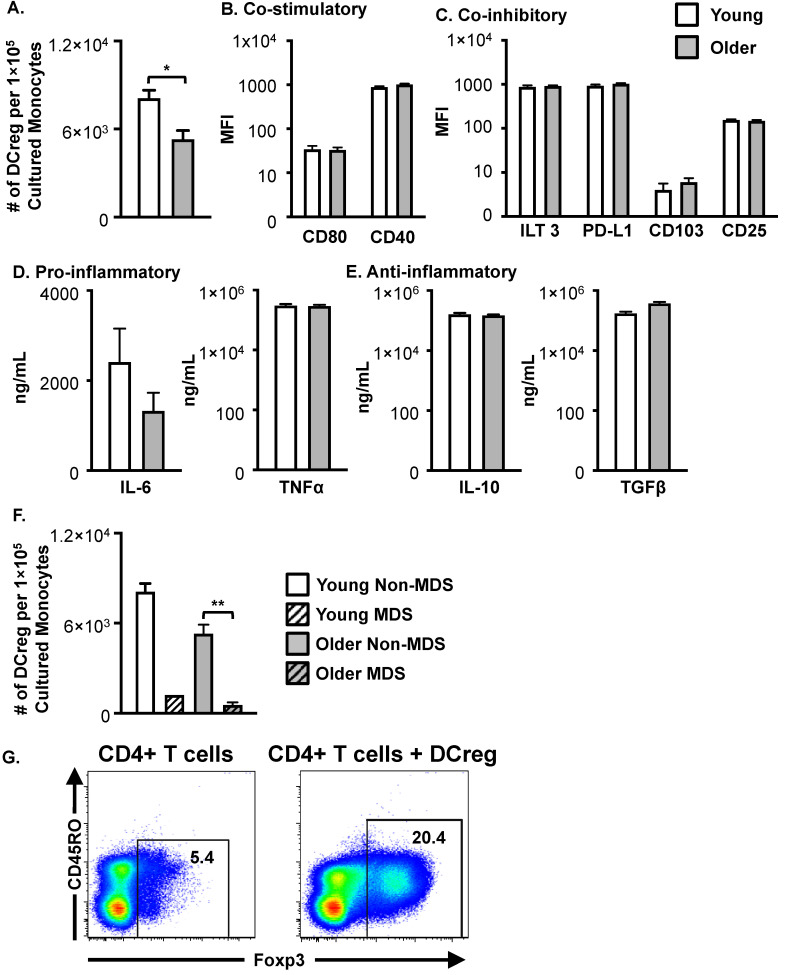
Older HSCT patient monocytes generate fewer DCregs than young HSCT patient monocytes yet maintain comparable phenotype and function. (**A**) Number of DCregs generated per 1 × 10^5^ cultured monocytes from young and older HSCT patients. Samples that did not grow were excluded. Young *n* = 11; Older *n* = 26. HSCT patient DCreg expression of (**B**) co-stimulatory molecules and (**C**) co-inhibitory molecules. Young *N* = 10; Older *N* = 23. (**D**) Pro- and anti- (**E**) inflammatory cytokine production by DCregs generated from young and older HSCT patients. IL-6, TNFα, and TGFβ ELISAs, *n* = 7 young; *n* = 16 older. IL-10 production exceeded maximal standard in 3 young and 7 older DCreg cultures (exceeded 2 × 10^5^ ng/mL IL-10) and were not included (*n* = 4 young; *n* = 9). (**F**) Young and older MDS patient DCreg numbers. Young non-MDS *n* = 11; Young MDS *n* = 1; Older non-MDS *n* = 24; Older MDS *n* = 7. Statistical comparison between young and older MDS DCreg numbers was not performed due to insufficient sample size. ** = *p* < 0.01. (**G**) DCreg generated from healthy monocytes induces Foxp3+ Treg. Representative plots are shown. *n* = 3. Data in A-F are mean ± SEM. * = *p* < 0.05.

## Data Availability

The data presented in this study are available upon request from the corresponding author.

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
