# Peer review of "Generation of Human Regulatory Dendritic Cells from Cryopreserved Healthy Donor Cells and Hematopoietic Stem Cell Transplant Recipients"

_cells, 2023, doi:10.3390/cells12192372_

Round 1

Reviewer 1 Report

Dear Editor,

The manuscript entitled “Generation of Human Regulatory Dendritic Cells from Cryo-preserved Healthy Donor Cells and Hematopoietic Stem Cell Transplant Recipients” explored cryopreservation effects on monocyte derived-regulated dendritic cells applicable on GVHD which could be utilized in patients. The study was well-designed and the results were well-discussed. However, there remains some issues to be discussed:

1.      There is some inconsistencies in writing: hematopoietic stem cell transplant or transplantation (HSCT)?, “morbidity and morbidity”, monoctes, α and β types missed in TNF and TGF in many places of manuscript, respectively.

2.      All of the methods need referencing.

3.      Histopaque density properties and cat. No is required.

4.      Regarding Histopaque density gradient medium, the manufacturer mentions: To the best of our knowledge, the chemical, physical, and toxicological properties have not been thoroughly investigated.” Therefore, it is not a GMP product and the epigenetic or long-term carcinogenic effects of the medium have not been investigated. Is it ethical to use this product on patients without investigating the long-term side effects?

5.      Why was letter x used instead of the multiplication sign in presenting cell numbers?

6.      There are numerous misspellings in showing exponential symbols in cell numbers.

7.      It’s better to describe manufacturers in the first mentioning (e.g., X-vivo media)

8.      For avoiding reader confusion, dots should be removed from hours and days symbols (d. and hr.)

9.      Using huge amounts of (50 mg/mL) of TGF-β1, GM-CSF, IL-4, and IL-10 is not normal. It is required to give the related references.

10.  Manufacturer and cat. no. for anti-CD16/32 was missed.

11.  Cat no. and fluorochrome type for each antibody is required.

12.  It is required to give the reference for highly diluted (0.1%) formaldehyde used for fixation.

13.  In line 125, it’s suggested to use co-cultured instead of cultured.

14.  Results need to be divided into subsections.

15.  In line 161, “levels of co-stimulatory (Fig.1B) and inhibitory molecule”, the source of molecules? in the culture medium? How long after culturing? should be added.

16.  In figure 1., what is the reason for low efficiency of DCreg generation from monocytes (~ 10%)? Differentiation efficiency in other references is required to compare. Cryopreservation timing length? young or old samples?

17.  In figure 1 and 2, cultured monocytes instead of “monocytes cultured” is suggestible.

18.  In line 231, lesser extent compared to what?

19.  Supplementary Fig. 4:"Young and older HSCT patient generated DCreg express low levels of HLA-DR and CD11c" Compared to what? CDc?

20.  As Supplementary figure 5. shows, there is high variation between cell viability and deterioration related to disease type. Hence, it is required to assess the number of DCreg generation per cultured monocytes in each disease and compare them by analyzing statistically.

21.  In line 289, GVT?

22.  In line 326, IDO?

Reviewer 2 Report

In this study the Authors analyzed the effect of cryopreservation, and the effect of donor's age on DCreg generation. They tested the influence of both age and cryopreservation in healthy donor and HSCT recipients. Crypreservation and age of healthy donors did not affect the number and characteristics of DCreg generated (co-stimulatory or inhibitory molecule expression, cytokine release), while older patients, and MDS patients  produced significantly fewer DCreg.

The results of this study are relevant to of the hypothetical implication in use of DCs for prevention/treatment of GVHD, as described in detail by the Authors.

There are minor issues:

- The group of patients analyzed is heterogeous (different diseases and different stages of disease) to draw meaningful conclusions. It is expected that hematological diseases may affect the performance in DCreg production, however this finding should be verified in a more homogeneous population of patients (e.g. AML in remission, or MDS). 

- in the discussion the Authors should comment that their results, although relevant and intriguing, they need to be confirmed in a larger series of patients/donors and in more experiments.

- line 326: IDO is not defined;

- line 16: the reference [16] is not appropriate;

- lines 367-369: the text is not clear, there are errors.

Round 2

Reviewer 1 Report

Dear Editor,

The revised version of the manuscript is improved and meets required criteria for publication.